# The Effect of Hypothermia and Osmotic Shock on the Electrocardiogram of Adult Zebrafish

**DOI:** 10.3390/biology11040603

**Published:** 2022-04-15

**Authors:** Elodie Arel, Laura Rolland, Jérôme Thireau, Angelo Giovanni Torrente, Emilie Bechard, Jamie Bride, Chris Jopling, Marie Demion, Jean-Yves Le Guennec

**Affiliations:** 1PhyMedExp, Université de Montpellier, Inserm U1046, UMR CNRS 9412, 34090 Montpellier, France; elodie.arel@inserm.fr (E.A.); jerome.thireau@inserm.fr (J.T.); emilie.bechard@inserm.fr (E.B.); jamie.bride@inserm.fr (J.B.); marie.demion@inserm.fr (M.D.); jean-yves.le-guennec@inserm.fr (J.-Y.L.G.); 2Institut de génomique fonctionnelle (IGF), Université de Montpellier, Inserm U1191, UMR CNRS 5203, 34094 Montpellier, France; laura.rolland@igf.cnrs.fr (L.R.); angelo.torrente@igf.cnrs.fr (A.G.T.)

**Keywords:** electrocardiogram, hyperosmotic therapy, hypothermic therapy, longitudinal studies, Bazett’s formula, zebrafish

## Abstract

**Simple Summary:**

Assessing cardiac toxicity of new drugs is a requirement for their approval. One of the parameters which is carefully looked at is the QT interval, which is determined using an electrocardiogram (ECG). Before undertaking clinical trials using human patients, it is important to first perform pre-clinical tests using animal models. Zebrafish are widely used to study cardiac physiology and several reports suggest that although ECG measurement can be performed, the recording configuration appears to affect the results. Our research aimed to provide a comprehensive characterization of adult zebrafish ECG to determine the best practice for using this model during cardiac toxicity trials. We tested three recording configurations and determined that exposing the heart provided the most reliable and reproducible ECG recordings. We also determined the most accurate correction to apply to calculate the corrected QT, which makes the QT interval independent of the heart rate, a critical parameter when assessing drug cardiac toxicity. Overall, our study highlights the best conditions to record zebrafish ECG and demonstrates their utility for cardiac toxicity testing.

**Abstract:**

The use of zebrafish to explore cardiac physiology has been widely adopted within the scientific community. Whether this animal model can be used to determine drug cardiac toxicity via electrocardiogram (ECG) analysis is still an ongoing question. Several reports indicate that the recording configuration severely affects the ECG waveforms and its derived-parameters, emphasizing the need for improved characterization. To address this problem, we recorded ECGs from adult zebrafish hearts in three different configurations (unexposed heart, exposed heart, and extracted heart) to identify the most reliable method to explore ECG recordings at baseline and in response to commonly used clinical therapies. We found that the exposed heart configuration provided the most reliable and reproducible ECG recordings of waveforms and intervals. We were unable to determine T wave morphology in unexposed hearts. In extracted hearts, ECG intervals were lengthened and P waves were unstable. However, in the exposed heart configuration, we were able to reliably record ECGs and subsequently establish the QT-RR relationship (Holzgrefe correction) in response to changes in heart rate.

## 1. Introduction

The electrocardiogram (ECG) is a widely used technique for analyzing the electrical activity of the heart and provides a wealth of information, enabling clinicians to determine discrepancies in patients suffering from a variety of different cardiomyopathies. ECG is also a standard control used during drug development and current legislation stipulates strict ECG criteria which must be met before a novel drug can be approved. The typical ECG consists of three main components, the P wave, which represents the depolarization of the atria, the QRS complex, which represents the depolarization of the ventricles occurring during systole, and the T wave, which represents the ventricular repolarization occurring during diastole (Figure 1A,B). One commonly assessed ECG parameter is the QT interval (the time interval between the start of QRS complex (ventricular depolarization) and the end of the T wave (ventricular repolarization)). Prolongation of the QT interval may result in fatal arrhythmias, which is why this parameter is a standard measurement when testing novel drug cardiotoxicity. New drugs must not significantly affect the QT interval in order for them to be approved for use in humans. Because the QT interval is inherently linked to the heart rate, it is standard practice to correct the QT interval to take into account changes/differences in heart rate which are unrelated to the QT interval. Such changes would otherwise confound this measurement (for example someone with a slow heart rate will have a longer uncorrected QT interval than someone with a fast heart rate, when they are in fact the same), resulting in the corrected QT or QTc. Prior to expensive clinical trials, it is important to first assess cardiotoxicity in animal models such as zebrafish. Zebrafish cardiac physiology is highly comparable with humans and in this respect, the zebrafish has become a powerful tool for determining the cardiotoxicity of novel pharmacological agents [1,2,3,4,5,6]. In particular, humans and zebrafish both exhibit similar ECGs [1,2,7,8] with discernable P waves, QRS complexes, and T waves.

The first ECG recordings in zebrafish were obtained by Milan et al. [1]. Since that paper, many papers have been released describing different methods to record ECG in zebrafish, but none were dedicated to comparing different methods to evaluate the plus-value of a given method for a given set of experiments. In this study, we aimed to evaluate and compare ECG profiles, measured longitudinally on the same animal, in three different configurations (unexposed heart, exposed heart, and isolated heart) in order to determine which technique is the most accurate for longitudinal studies. Furthermore, we endeavored to characterize the QT-RR relationship in each configuration in order to determine which of these techniques is the most accurate and reliable for assessing the QTc interval. To this end, we also sought to determine what effect clinical therapeutic treatments such as hyperosmolality (employed during traumatic brain injury treatment) and hypothermia/targeted temperature management (employed during stroke/heart attack treatment) had on ECG recordings in the three different configurations. Because these treatments are often combined with medication, we sought to determine whether adult zebrafish can be used as a model for testing drug cardiotoxicity, which could be utilized in combination with these therapeutic treatments.

## 2. Materials and Methods

### 2.1. Zebrafish Strains and Husbandry

Zebrafish were maintained under standardized conditions and experiments were conducted in accordance with local approval (APAFIS#2021021117336492 v5) and the European Communities council directive 2010/63/EU. All experiments were performed on 6–8 month old AB wildtype fish.

### 2.2. ECG Recording

Zebrafish were anesthetized in tricaine (160 mg/L). Then, they were placed ventral side up in a slit sponge. Two 29-gauge stainless steel micro-electrodes (MLA1213, AD Instruments, Dunedin, Otago, New Zealand) were positioned along the ventral midline. The reference electrode was either placed in the bath or out, depending on what created the clearest signal. In the unexposed heart configuration, the positive electrode was positioned just above the heart and the negative electrode in front of the anal fin to record ECG. In the exposed heart configuration, we surgically opened the cardiac cavity, and the electrodes were positioned close to, but not touching, the cardiac muscle. Lastly, in the extracted heart configuration, the heart was extracted under a dissection microscope and the electrodes were positioned in the main axis of the ventricle. In this configuration, the heart activity was stable during the ECG recording. In all conditions and in accordance with the 3Rs, ECGs were recorded for exactly one minute only. After recording ECGs in the unexposed heart configuration, individual fish were returned to their tanks and maintained separately for few days before recording the ECG again in the exposed heart configuration. No treatment was applied after surgery, in order to avoid affecting the ECG recordings. Individual fish were again allowed to recover for a few days in the system before recording ECGs in the terminal extracted heart configuration. In the three configurations, ECG signals were amplified and digitized using a BioAmp (FE231, AD Instruments) and a PowerLab (16/35, AD Instruments). Data were subsequently processed using LabChart Pro v8 Software (AD Instruments, Dunedin, Otago, NZ) and the ECG analysis module (AD Instruments, Dunedin, Otago, NZ). Recordings were made in the range 0–10 mV. A 50 Hz notch digital filter was then applied, and a sliding averaging algorithm provided by the software was used to smooth the traces. To reduce the biological variations, the same animals were used in the three different configurations of ECG measurements. A descriptive schematic of our equipment is provided in supplemental Appendix A and schematic representations of the different configurations appear in Figure 1C–E.

### 2.3. QT Analysis and Correction

ECG were manually analyzed as auto-identification waveforms are often associated with cursor placement errors. The most common software error was a failure to distinguish between P and R waves, resulting from excessive noise or the instability of the isoelectric line. As a result, the software also failed to automatically and reliably identify the Q wave. Thus, all ECG waveforms were systematically identified and the errors in positioning the PQRST cursors were fixed by the operator to avoid misinterpretation. We chose to use the Ppeak, Qpeak, Speak, and Tpeak to analyze the ECG instead of the beginning/end of the waveforms in order to reduce both the risk of misplacement of the wave and the inter-operator dependence of the results. The PR interval was estimated as the time between the peak P and the peak R waves and the QT interval as the time between the peak Q and peak T waves (Figure 1A). Due to interspecies variability of the QT-RR relationship, we used the corrected QT (QTc) formula described by Holzgrefe et al. (2014) [9]:$${\mathrm{QT}}_{c}=\frac{\mathrm{QT}}{{\left(\frac{\mathrm{RRraw}}{\mathrm{RRref}}\right)}^{s}}$$
where s is the slope of the linear relationship Log(QT) = f(Log(RR)) and RRref is the reference RR. The beating frequency is around 120 beats per minute, thus RRref is equal to 0.5 s [1,3].

### 2.4. Solutions and Experimental Protocols

Isolated hearts were kept in a Tyrode solution: NaCl 140 mM, KCl 5 mM, CaCl_2_ 2 mM, MgCl_2_ 1mM, HEPES 10 mM, pH 7.4 (309 mosm.L^−1^). To determine the effects of reducing the temperature, the Tyrode solution was chilled from 28 °C to 5 °C. One-minute ECG recordings were obtained in warm Tyrode (control, 28 °C), cold Tyrode (5 °C), and warm Tyrode as a washout condition. 

To produce an osmotic shock, we first prepared an isotonic hypo-Na Tyrode solution: NaCl 90 mM, mannose 100 mM, KCl 5 mM, CaCl_2_ 2 mM, MgCl_2_ 1 mM, HEPES 10 mM, pH 7.4 (309 mosm.L^−1^). We then made a hyposmotic solution: NaCl 90 mM, KCl 5 mM, CaCl_2_ 2 mM, MgCl_2_ 1 mM, HEPES 10 mM, pH 7.4 (209 mosm.L^−1^). Finally, we prepared a hyperosmotic solution: NaCl 90 mM, mannose 200 mM, KCl 5 mM, CaCl_2_ 2 mM, MgCl_2_ 1 mM, HEPES 10 mM, pH 7.4 (409 mosm.L^−1^). The ECGs were recorded in all conditions following this sequence: Tyrode–Isotonic–Hyposmotic–Isotonic–Hyperosmotic–Isotonic.

For both the unexposed and exposed heart configurations, we first removed 1 mL of the bath solution, then 1mL of the osmotic shock solution, or 1ml of the hypothermic solution was applied onto the cardiac area prior to measurement.

### 2.5. Statistical Analysis

All data were processed in GraphPad Prism 9.2.0. Data (GraphPad, San Diego, CA, USA) are expressed as mean (SEM). The parameters measured (PR, QRS and QT intervals, HR, and QTc) were compared between the different recording configurations first and then within each configuration for the temperature and osmotic protocols. For each recording, PR and QT intervals were excluded when P or T waves were only detected in 15% or less of the ECG complexes. We used repeated measures one-way ANOVA followed by the Tukey’s post hoc test to compare 3 or more normal groups, or the Kruskal-Wallis test followed by Dunn’s post hoc test to compare 3 or more groups which did not pass normality testing. A significant difference is labeled as follow: *p* < 0.05 *, *p* < 0.01 **, *p* < 0.001 ***, *p* < 0.0001 ****.

## 3. Results

### 3.1. Basal Characteristics of Adult Zebrafish ECG

In order to determine the most reliable method for analyzing the QT interval in adult zebrafish we recorded ECGs in three different configurations, unexposed heart, exposed heart, and extracted heart (Figure 1C–E). From these recordings, we were able to determine the average PR interval, QRS duration, RR interval, heart rate, QT interval, and calculate the QTc interval (Figure 1A and Table 1). Our analysis indicates that there is no significant difference in heart rate between the three configurations (Table 1). When we compared the two in vivo configurations (unexposed heart and exposed heart), we found that the QRS interval was longer in the exposed heart configuration suggesting a possible slowing of ventricular conduction. However, assessing the QT interval in the unexposed heart configuration presented difficulties, as we were only able to detect the T wave in 8 out of 20 samples (Table 1 and Table 2, Appendix A). Under normal conditions in humans, T waves are primarily positive, however, although negative (inverted) T waves are associated with a number of cardiomyopathies in adult humans, they are in fact predominant in children [10]. Our data indicates that in adult zebrafish, the majority of T waves are negative (albeit well below the threshold of what is considered abnormal for an adult human) (Figure 1C–E and Table 2), which may reflect differences in cardiac anatomy between humans and adult zebrafish. Taken together, our data indicate that the most reliable technique for assessing the QT interval accurately is the exposed heart configuration.

### 3.2. Effects of Osmotic Shock on Adult Zebrafish ECG Characteristics

In humans, cerebral edema resulting from brain injury is frequently treated with hyperosmotic therapy to relieve inter cranial hypertension [11]. However, recent evidence suggests that elevating plasma osmolality can also lead to increased QTc and a higher risk of cardiac arrhythmias [12,13]. We therefore sought to determine what effect osmotic perturbations had on zebrafish ECG patterns. To achieve this, we recorded ECGs in each of the three configurations in either hyperosmotic, isosmotic, or hyposmotic conditions. 

Interestingly, osmotic challenge did not significantly alter the ECG characteristics in any of the three configurations when compared to the isosmotic controls (Table 3). However, despite the lack of significant differences in the ECG characteristics we were able to determine a clear positive QT-RR relationship in the exposed heart configuration, which was not present in the unexposed configuration (Figure 2A,B).

This positive correlation was also observed in the extracted heart configuration, however there was also a much higher variation of QT interval in relation to increasing RR in this configuration when compared to the exposed heart recordings (Figure 2B,C). Taken together, our data indicate that although adult zebrafish can be a useful model for testing drug cardiotoxicity, due diligence should be taken when assessing therapies which can affect plasma osmolality, as these may not affect zebrafish in the same way as humans. However, it also appears that, unlike rodents, zebrafish hearts (like humans) elicit a positive QT-RR relationship, allowing the QTc interval to be calculated which is vital when screening novel pharmaceuticals.

### 3.3. Effects of Hypothermia on Adult Zebrafish ECG Characteristics

Therapeutic hypothermia (also known as targeted temperature management) is often employed as neuroprotection in patients who have suffered a cardiac arrest or form other ischemic episodes resulting in reduced blood flow in the brain [14,15]. In order to ascertain whether zebrafish hearts respond to hypothermic conditions in a similar manner to humans, we recorded ECGs in each configuration using either a control, warm (28 °C) tyrode solution or a chilled (5 °C) solution.

In the unexposed heart configuration, there was a significant reduction in the heart rate accompanied by significant elongations in the PR interval and QRS complex (Table 4). Although hypothermic conditions resulted in a lower heart rate in the exposed heart configuration, this was below the threshold of significance. However, although the PR interval increased, the QRS complex remained unchanged (Table 4). Importantly, as is the case in humans, hypothermia resulted in a significant lengthening of the QT interval in the exposed heart configuration (Table 4), which was also the case for the extracted heart configuration. No significant differences were observed between the warm (control) and the warm (washout) conditions, indicating that these effects are all reversible (Table 4). Interestingly, hypothermia appears to increase the QT interval independently of the RR interval in both the exposed and extracted heart configurations (Figure 3), indicating that this treatment has a direct effect on the QT interval. Taken together, our data indicate that the exposed heart configuration appears to be the most reliable technique when using zebrafish to test hypothermic therapeutic treatments.

### 3.4. QT-RR Relationship

The first description of QTc dates back to 1920 with Bazett’s work, which remains the most frequently used formula [16]. Calculating the QTc is paramount during any drug cardiotoxicity testing. To achieve this, it is necessary to establish the QT-RR relationship and in this sense identifying a clear T wave is essential. In the exposed heart configuration, it is relatively easy to identify the T wave and thus build the QT-RR relationship. To characterize this relationship, we decided to use the Holzgrefe formula since it can be applied to different animal species if the Log(QT)–Log(RR) relationship is linear (QTch) [9]. When this is applied to the data obtained from the osmotic shock analysis, we could observe a linear relationship with a slope of 0.2064 (Figure 4A). Based on this data, we were subsequently able to establish the non-linear fitting of the QT-RR relationship (Figure 4B). Using Holzgrefe’s formula, we calculated the QTch (corrected Holzgefe) and subsequently plotted the QTch-RR relationship (Figure 4C). It is apparent that there is no correlation between the QTch and the RR interval, showing that QTch is independent of the RR interval as expected for a good correction (Figure 4C). Conversely, if we apply Bazett’s correction formula (QTcb) to our data, this transforms the positive QT-RR relationship into a negative QTcb-RR relationship (Figure 4C). Using this calculation, the QTcb is exaggeratedly increased by tachycardia (increased heart rate) and decreased by bradycardia (decreased heart rate), indicating that Holzgrefe’s formula provides the most accurate QTc value. To confirm that this was not due to our data, we re-analyzed previously published ECG recordings [1] and fitted them with either the Holzgrefe or Bazett formula (Figure 4D). In this manner, we were able to determine that applying the Holzgrefe formula results in a QTch that is independent of the RR interval, as observed with our own dataset. Conversely, applying the Bazett formula results in biased data similar to that which we observed with our own ECG recordings (Figure 4D). Taken together, our data indicate that the exposed heart configuration is the most reliable for identifying T waves and establishing the QT-RR relationship. Furthermore, it is also apparent that Holzgrefe’s formula appears to be the most accurate method for calculating the corrected QT interval.

## 4. Discussion

The first ECG recordings in zebrafish were obtained by Milan et al. [1]. Subsequently, several research groups have tried to adapt the recording techniques and/or analysis methods to improve the quality and reliability of the signal. Despite some improvements, Liu et al. [4] highlighted discrepancies between the ECG signals obtained from different research groups regardless of their recording methods. Currently, there are two different techniques used by research teams to record ECG in adult zebrafish hearts. The first is non-invasive and involves positioning the electrodes on the body surface [17,18,19], while the second technique involves inserting the electrodes either 1 mm into the dermis (i.e., in the pectoral muscles) or directly onto the surface of an exposed heart [1,3,4,5,20,21,22,23,24,25,26]. It is apparent that the choice of technique has a major influence on the raw ECG signal. For example, by using the non-invasive technique it appears that Q, S, and T waves are difficult to identify [17,19], and are generally assigned manually after the raw data has been processed [18]. In contrast, by using electrodes that allow direct access to the hearts electrical activity it is possible to reliably identify P waves, the Q, R, S complex, and T waves [2,3,4,5,23,26,27]. Furthermore, post-experimental mathematical processing can also be used to reduce background noise [25,28,29,30]. However, it should be noted that T waves generated during the repolarization of the ventricle are difficult to discriminate using either technique. T wave analysis is an important factor in determining the QT interval during cardiotoxicity testing, and in this sense it is important to be able to reliably and accurately measure this parameter. At present, there appears to be confounding data regarding the nature of T waves in the adult zebrafish heart. For example, although several studies have reported negative T waves [1,18,22,31], other groups have actually recorded positive T waves [2,3,4,19,20,21,23,26,27]. It should be noted that in some cases, the T wave appears to be rather difficult to identify distinctly when the signal is recorded using a microelectrode array [24,25,30]. Lastly, ECGs can also be recorded on isolated hearts [32,33]. However, in this configuration the interpretation of the results is confounded by the absence of autonomic regulation by the nervous system [33,34,35]. This is particularly relevant, as recent evidence suggests that the cardiotoxicity of certain drugs linked to ventricular repolarization involves autonomic dysregulation [36,37]. Furthermore, adult zebrafish are also widely used to study cardiac regeneration using methods such as cryoinjury and ventricular resection [22,38]. In humans, such injuries i.e., myocardial infarction, are characterized by electrophysiological alterations, which can be detected using ECG. Thus, the techniques we describe here can also be used to explore the electrophysiological changes associated with cardiac regeneration [21,22,27]. Zebrafish are being increasingly employed to understand more about congenital cardiac arrhythmias such as Holt-Oram syndrome [39]. While many of these studies perform ECG on zebrafish larvae, it would be beneficial to also study the ECG of a fully developed heart in an adult zebrafish. For example, one can imagine using a larval cardiac arrhythmia model in a large scale screen for novel antiarrhythmic compounds. Once positive hits have been identified, it will be beneficial to employ the ECG techniques we describe in order to assess the effects these drugs have on the ECG of adult zebrafish from the same cardiac arrhythmia line. 

In this study, we have compared the ECG characteristics in three different configurations in order to evaluate their limitations and advantages. We have also assessed the effects of osmotic shock and hypothermia, two frequently used clinical treatments which are known to affect cardiac electrophysiology. By analyzing these data, we have been able to better delineate the QT-RR relationship, which is critical in calculating the QTc. 

It is apparent from our own (and others’ [17,18]) research that recording adult zebrafish T waves is inherently difficult and susceptible to operator bias. Indeed, in the unexposed heart configuration, we could barely detect any T waves at all. This is in contrast to the exposed heart configuration, where we could readily observe distinct T waves and thus calculate the QT interval. Our findings are in agreement with Liu et al. [4] who also found that removing the dermis and opening the pericardium significantly improved T wave detection. Interestingly, adult zebrafish T waves can be either positive, as in healthy humans, or negative, as is in young children or other species such as canines. This phenomenon has also been previously described by Tsai et al. [32], who could detect a mixture of positive T waves (45%) and negative T waves (25%) (30% undetectable). From our own analysis in the exposed heart configuration (as opposed to the extracted heart configuration used by Tsai et al.), we observed similar levels of undetectable T waves (25%), however we found positive T waves in 20% of the recordings and negative T waves in 55% of the recordings. Furthermore, we observed that while the polarity of the T wave can be different between individual zebrafish, it can also change within the same zebrafish during the experiment. This might be explained by the changing propagation of electrical gradient [20].

In humans, hyperosmotic therapy is frequently used during the treatment of traumatic brain injuries. Recent evidence suggests that this treatment can also adversely affect the QT interval, which could lead to potentially fatal cardiac arrhythmias. Our data indicate that, unlike humans, zebrafish cardiac electrophysiology is highly tolerant to changes in osmolality. Because zebrafish live in freshwater fish, they are constantly subjected to the osmotic gradient between their environment and interstitial fluids. To prevent osmotic damage, zebrafish are able to rapidly regulate Na+ and Cl− transport [40]. For example, a recent study from Kennard et al., showed that in zebrafish, the process of wound healing was not affected by changes in osmolality despite considerable cell swelling [41]. The ability to adapt to osmotic shock, even after disrupting the epidermal barrier, was confirmed by our own results. This feature should also be taken into account when using zebrafish to perform cardiotoxicity tests of novel pharmaceuticals, which may affect plasma osmolality, or which may be used in combination with hyperosmotic therapy.

Following serious health complications, such as a heart attack or stroke, which result in a drastically reduced blood supply to the brain, target temperature management is often employed to reduce the risk of neuronal damage [42]. However this treatment is also known to impact cardiac electrophysiology and can lead to the lengthening of the QT interval and the potential for lethal arrhythmias [14,42]. Similarly, using the exposed heart configuration, we also observed that subjecting zebrafish to hypothermia resulted in a significant increase in the QT interval. This indicates that zebrafish would be an excellent model for testing the cardiotoxicity of novel therapies which cause or are used in conjunction with hypothermia.

Current legislation stipulates that before a new drug can be approved it must be demonstrated that it does not significantly alter the QT interval in human subjects. Clinical trials are very costly and before they can be undertaken, pre-clinical assessment using animal models must be performed. The adult zebrafish is an excellent model for cardiotoxicity tests; however, it is imperative to first establish the QT-RR relationship accurately in order to ascertain whether any particular treatment actually affects the QT interval. The QT interval varies depending on the RR interval [1,2]. Thus, it is important to correct the QT and make it independent of the heart rate when studying experimental conditions that change the RR. The standard equation used to correct the QT (QTc) is ${\mathrm{QT}}_{c}=\frac{\mathrm{QT}}{{\mathrm{RR}}^{s}}$. In humans, where s equals 0.5 (Bazett’s equation) or 0.33 (Fridericia’s equation). These slope values have been widely used in correction formulae when studying zebrafish ECG recordings [4,5,20,21,22,31,32,43]. However, it is vitally important to consider that the adult zebrafish mean cardiac frequency is around 120 bpm, while in humans it is around 60 bpm. In this respect, we divided the RR interval by a zebrafish reference RR interval of 0.5 s [9]. This correction allowed us to accurately calculate the QTc independently of the RR interval, something we could not achieve when employing Bazett’s formula.

Finally, the analysis of the QT-RR relationship raises some interesting questions. Indeed, when compared to previous reports, we found a rather low s power of 0.18 (previously published s powers include 1.05 [1], 1.08 [3], 0.449 [44], and 0.58 [2]). This discrepancy between the QT-RR relationship maybe due to a number of factors such as the strain of fish [45], the experimental conditions, anesthesia, and other types of treatment. Recently, other confounding factors such as sex and body weight have also been described [46]. This suggests that the best protocol when studying zebrafish ECG is to first characterize the QT-RR of a given strain of fish in standardized experimental conditions before going further. We would also like to highlight the risk of bias induced by correction formula such as Bazett’s equation, which appears to be critical. Our data suggest that Holzgrefe’s formula provides the most accurate correction since the QTc does not change in relation to the RR. In contrast, Bazett’s equation produces a negative relationship. This is important because under conditions that can reduce the cardiac rate, it would appear that QTc interval would increase, which in fact would just be an artefact of the correction formula. In order to confirm this finding, we utilized previously published QT-RR data [1] and we applied either Holzgrefe’s correction formula or the Bazett formula. In this manner, we found that Bazett’s formula does not correct the QT, it even overestimates it at higher heart rates, while Holzgrefe’s formula perfectly corrected the QT. Thus, it is clear that Bazett’s formula has to be avoided to correct the QT interval in zebrafish. Furthermore, because Holzgrefe’s correction is made relative to the animals’ heart rate this allows the QTc to be calculated in the range of the measured QT, which is not the case without this normalization.

## 5. Conclusions

In conclusion, of the three commonly used techniques it seems that the exposed heart configuration provides the most reliable ECG recordings in adult zebrafish. Furthermore, despite the considerable evolutionary distance between zebrafish and humans, their ECG characteristics are remarkably similar, which highlights the utility of using adult zebrafish for cardiotoxicity tests over other animal models such as rodents.

## Figures and Tables

**Figure 1 biology-11-00603-f001:**
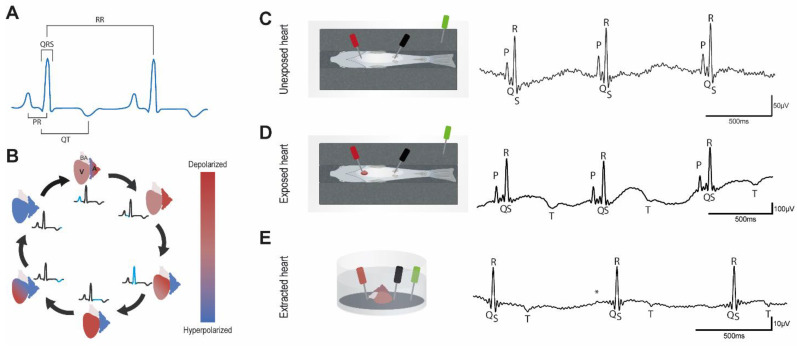
ECG representation and typical recordings; (**A**), Schematic representation of the interval measurement method. We chose to take the peak of the P and T waves to determine PR and QT intervals respectively to reduce the uncertainty of the measurements when taking before P waves or after T waves; (**B**), Cartoon representing the different depolarization and repolarization steps of the zebrafish heart cavities. A: atrium, V: ventricle, BA: bulbus arteriosus; (**C**–**E**), Schematic representation of the electrode positioning in the different configurations and representative traces; (**C**), Unexposed heart; (**D**), Exposed heart; (**E**), Extracted heart, * indicates putative P wave. Traces show three consecutive ECG complexes after application of a 50 Hz low-pass filter.

**Figure 2 biology-11-00603-f002:**
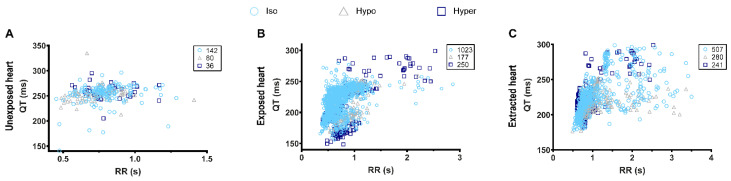
QT-RR relationship in different configurations of ECG measurement. (**A**), Unexposed heart; (**B**), exposed heart; (**C**), extracted heart. The number of individual QT-RR pairs, obtained in isosmotic, hyposmotic, and hyperosmotic conditions, is presented in the upper right corner of each graph and was obtain from *n* = 3 (unexposed heart), *n* = 10 (exposed heart), and *n* = 6 (extracted heart) adult zebrafish.

**Figure 3 biology-11-00603-f003:**
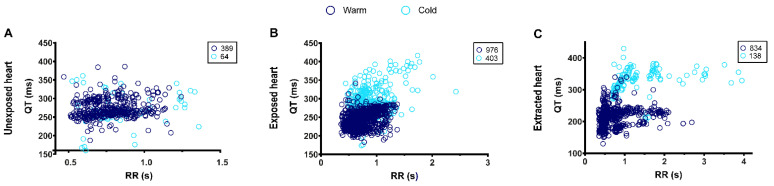
QT-RR relationship in different configurations of ECG measurement. (**A**), Unexposed heart; (**B**), exposed heart; (**C**), extracted heart. The number of individual QT-RR pairs, obtained in warm and cool conditions is presented in the upper right corner of each graph and was obtain from 5 (unexposed heart), 9 (exposed heart) and 7 (extracted heart) fish.

**Figure 4 biology-11-00603-f004:**
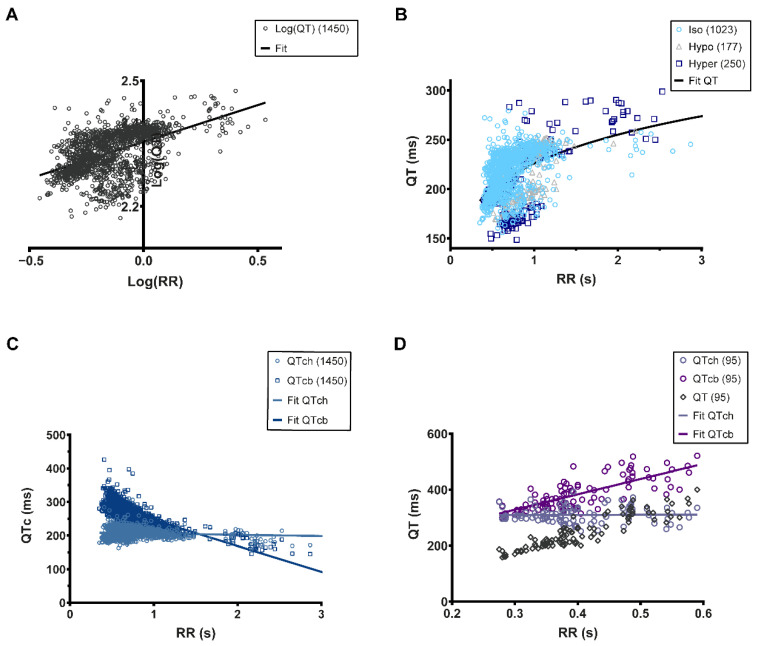
QT-RR relationship fitting. (**A**), Linear fitting (black line) of the Log(QT)–Log(RR) relationship). The slope of the relationship is 0.2064; (**B**), Non-linear fitting (black line) of the QT-RR relationship using the characteristics of the fitting shown in (**A**); (**C**), QTc-RR relationship using Holzgrefe’s correction formula (QTch) and Bazett’s equation (QTcb). These relationships were linearly fitted (blue-grey line: QTch; navy line: QTcb); (**D**), QT-RR relationship from the data published in Milan et al., 2006. The QT-RR raw data are in open black diamonds. The QT corrected by the Holzgrefe’s equation, QTch, are lilac opened circles, and the corresponding linear fitting is the lilac line, while the QT corrected by Bazett’s formula are opened purple circles and the corresponding linear fitting if the purple line. The number of individual QT-RR pairs is presented in the upper right corner of each graph.

**Table 1 biology-11-00603-t001:** The ECG characteristics for each recording configuration. Data are presented as mean (SEM). The sample size (*n*) is indicated in italic in brackets for each parameter. *p* values were calculated using either a one-way ANOVA(^A^) or the Kruskal-Wallis(^K^) test. * *p* < 0.05, ** *p* < 0.01.

Configuration	PR (ms) ^K^	QRS (ms) ^K^	QT (ms) ^K^	RR (ms) ^K^	HR (bpm) ^A^
Unexposed Heart	44.79 (1.90) *(20)*	25.93 (1.02) *(20)*	273.4 (7.65) *(8)*	766.8 (30.92) *(20)*	80.27 (3.07) *(20)*
Exposed Heart	41.44 (2.22) *(17)*	31.34 (1.54) * *(20)*	234.8 (6.29) *(16)*	699.9 (32.99) *(20)*	86.46 (4.40) *(20)*
Extracted Heart	49.87 (6.52) *(5)*	34.09 (4.94) *(17)*	213.4 (17.60) ** *(13)*	910.1 (85.80) *(18)*	74.68 (6.24) *(18)*

**Table 2 biology-11-00603-t002:** Characteristics of the T waves in the different configurations. Data are presented as percentages (proportion of animals indicated in brackets).

Configuration	Positive T Wave	Negative T Wave	Undetectable
Unexposed Heart	0% (0/20)	40% (8/20)	60% (12/20)
Exposed Heart	25% (5/20)	55% (11/20)	20% (4/20)
Extracted Heart	28% (5/18)	44% (8/18)	28% (5/18)

**Table 3 biology-11-00603-t003:** Effects of osmotic shock on the ECG characteristics. Data are presented as mean (SEM). The sample size (*n*) is indicated in italic in brackets for each parameter in the table. *p* values were calculated using either a one-way ANOVA (^A^) or the Kruskal-Wallis (^K^) test. NC: not calculated.

Unexposed Heart	PR (ms) ^K^	QRS (ms) ^A^	QT (ms) ^NC^	RR (ms) ^K^	HR (bpm) ^A^
**Tyrode**	46.15 (3.34)*(10)*	26.65 (1.24) *(10)*	271.5 (9.01) *(3)*	799.5 (41.08) *(10)*	76.82 (4.01) *(10)*
**Isosmotic**	57.01 (13.14) *(10)*	25.86 (1.45) *(10)*	265.1 (8.38) *(3)*	756.5 (44.95) *(10)*	81.34 (3.89) *(10)*
**Hyposmotic**	47.08 (2.33) *(9)*	26.11 (1.47) *(10)*	266.8 (4.24) *(3)*	727.7 (46.7) *(10)*	84.85 (4.25) *(10)*
**Wash isosmotic**	46.97 (1.95) *(7)*	24.86 (1.79) *(8)*	NC *(1)*	743.3 (38.04) *(8)*	82.27 (4.36) *(8)*
**Hyperosmotic**	46.59 (2.44) *(8)*	25.95 (1.22) *(10)*	NC *(2)*	766.2 (44.61) *(10)*	80.50 (4.26) *(10)*
**Exposed heart**	**PR (ms) ^A^**	**QRS (ms) ^A^**	**QT (ms) ^A^**	**RR (ms) ^A^**	**HR (bpm) ^A^**
**Tyrode**	43.78 (4.04) *(10)*	33.80 (1.46) *(10)*	235.0 (9.99) *(10)*	705.4 (45.32) *(10)*	88.48 (5.86) *(10)*
**Isosmotic**	42.13 (2.02) *(9)*	31.39 (1.47) *(10)*	241.3 (11.73) *(8)*	733.2 (149.0) *(10)*	85.3 (6.08) *(10)*
**Hyposmotic**	41.07 (2.02) *(9)*	29.27 (1.69) *(10)*	244.9 (12.9) *(7)*	716.6 (46.9) *(10)*	87.43 (6.37) *(10)*
**Wash isosmotic**	41.36 (1.98) *(9)*	29.39 (1.74) *(10)*	243.9 (14.02) *(6)*	710.4 (51.48) *(10)*	88.86 (6.92) *(10)*
**Hyperosmotic**	40.70 (1.83) *(9)*	31.07 (1.55) *(10)*	245.2 (14.48) *(5)*	729.6 (48.69) *(10)*	85.88 (6.18) *(10)*
**Extracted heart**	**PR (ms) ^NC^**	**QRS (ms) ^K^**	**QT (ms) ^A^**	**RR (ms) ^A^**	**HR (bpm) ^A^**
**Tyrode**	NC *(0)*	42.82 (8.06) *(9)*	233.9 (19.11) *(6)*	954.0 (102.5) *(10)*	69.79 (7.80) *(10)*
**Isosmotic**	NC *(0)*	33.23 (5.01) *(9)*	232.2 (6.18) *(5)*	1239.0 (240.0) *(10)*	67.24 (12.14) *(10)*
**Hyposmotic**	NC *(0)*	32.00 (4.84) *(9)*	224.4 (6.71) *(7)*	1201.0 (202.6) *(10)*	63.27 (9.95) *(10)*
**Wash isosmotic**	NC *(0)*	31.99 (5.05) *(9)*	215.7 (6.17) *(7)*	980.7 (180.7) *(10)*	77.25 (10.31) *(10)*
**Hyperosmotic**	NC *(0)*	31.18 (4.14) *(8)*	227.7 (8.44) *(7)*	1132.0 (186.0) *(10)*	64.46 (8.23) *(10)*

**Table 4 biology-11-00603-t004:** Effects of temperature on the characteristics of the ECG. Data are presented as mean (SEM). The sample size (*n*) is indicated in italic in brackets for each parameter in the table. *P* values were calculated using either a one-way ANOVA(^A^) or the Kruskal-Wallis(^K^) test. * *p* < 0.05, ** *p* < 0.01, **** *p* < 0.0001.

Unexposed Heart	PR (ms) ^A^	QRS (ms) ^A^	QT (ms) ^K^	RR (ms) ^A^	HR (bpm) ^A^
Warm (control)	43.42 (1.90) *(10)*	25.22 (1.66) *(10)*	273.4 (7.65) *(5)*	734 (45.94) *(10)*	83.72 (4.60) *(10)*
Cold	78.47 (6.87) **** *(10)*	37.59 (2.71) ** *(10)*	252.2 (43.08) *(2)*	1401 (196.9) ** *(10)*	50.86 (7.30) ** *(10)*
Warm (washout)	46.95 (2.23) *(8)*	25.33 (2.11) *(9)*	272.5 (26.66) *(3)*	921.4 (81.02) (10)	69.05 (5.84) *(10)*
Exposed heart	**PR (ms) ^A^**	**QRS (ms) ^A^**	**QT (ms) ^A^**	**RR (ms) ^A^**	**HR (bpm) ^A^**
Warm (control)	38.48 (1.91) *(8)*	28.88 (2.55) *(10)*	234.5 (4.07) *(6)*	694.4 (50.34) *(10)*	84.44 (6.80) *(10)*
Cold	49.78 (2.96) ** *(9)*	29.96 (2.35) *(10)*	292.6 (11.86) ** *(9)*	956.6 (76.04) * *(10)*	65.36 (4.43) *(10)*
Warm (washout)	41.32 (1.94) *(9)*	25.99 (2.96) *(10)*	244.5 (6.57) *(8)*	762.9 (74.31) *(10)*	83.74 (76) *(10)*
Extracted heart	**PR (ms) ^A^**	**QRS (ms) ^K^**	**QT (ms) ^K^**	**RR (ms) ^K^**	**HR (bpm) ^A^**
Warm (control)	49.87 (6.52) *(5)*	24.26 (3) *(8)*	195.8 (27.90) *(7)*	855.1 (150.3) *(8)*	80.79 (10.29) *(8)*
Cold	56.23 (6.65) *(4)*	38.80 (3.92) *(8)*	341.8 (9.05) ^*^ *(5)*	2038 (462.5) ^*^ *(8)*	39.28 (6.97) ^*^ *(8)*
Warm (washout)	41.42 (3.45) *(3)*	34.52 (11.95) *(8)*	220.5 (11.62) *(7)*	846 (116.9) *(8)*	77.68 (10.08) (8)

## Data Availability

Not applicable.

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
