# Peer review of "The Effect of Hypothermia and Osmotic Shock on the Electrocardiogram of Adult Zebrafish"

_biology, 2022, doi:10.3390/biology11040603_

Round 1
Reviewer 1 Report
The manuscript by Arel and co-workers describe the ECG recording conditions in adult zebrafish and compares 3 recording configurations. By reading the manuscript my recuring feeling was that the authors concentrated on a narrow segment of existing literature and missed numerous key findings. Likely that was the reason of the claim that in general T wave recording is difficult in zebrafish. Using a similar configuration (LabChart) for example Dhillon and co-workers have published ECG recordings in zebrafish larva where T wave were measured with the same software (Plos One, 2013).
Below I list several additional concerns about the manuscript:
- Corrected QT has been described much earlier than the cited 2014 Holzgrefe paper. The authors should find the original article back in the 1950’s (or earlier) where it was first used.
- Another problem with the manuscript is the comparison of QTc and RR. QTc is the QT interval corrected for the heart rate. Since heart rate is the RR interval in milliseconds, comparing QTc to RR must be linear, and makes not much sense to plot it.
- The electrode position determines whether the T wave has a positive or negative deflection (check for example Crowcombe et al 2016 (PlosOne)). It needs to be discussed.
- I assume that the RR intervals listed on the tables are in milliseconds not seconds.
Throughout the manuscript reading I asked myself, why the authors recorded from adult zebrafish that required animal permit and seemingly made the recording difficult. It is much easier to do ECG recording from zebrafish larvae that needs no animal permit, can be done in medium to high throughput and pharma companies use it for drug discovery experiments rather than adult zebrafish.
However, there are merits of the manuscript. For example, the experiment to gain insight to the effect of hypothermia ECG is worth expanding and further exploring, and I suggest taking the hypothermia experiment and publish it separately.
Author Response
Reviewer 1
The manuscript by Arel and co-workers describe the ECG recording conditions in adult zebrafish and compares 3 recording configurations. By reading the manuscript my recuring feeling was that the authors concentrated on a narrow segment of existing literature and missed numerous key findings. Likely that was the reason of the claim that in general T wave recording is difficult in zebrafish. Using a similar configuration (LabChart) for example Dhillon and co-workers have published ECG recordings in zebrafish larva where T wave were measured with the same software (Plos One, 2013).
Authors’ response: We thank the reviewer for this insightful comment. As we are sure the reviewer is aware, the main differences between the work presented by Dhillon et al. (2013) and our study is that Dhillon et al used zebrafish larvae to record the ECG whereas we used adults. We have subsequently amended the title of our manuscript to indicate that our study has been performed on adult zebrafish. We believe we have given an accurate account of the current state of the heart regarding ECG measurements in adult zebrafish.
Below I list several additional concerns about the manuscript:
- Corrected QT has been described much earlier than the cited 2014 Holzgrefe paper. The authors should find the original article back in the 1950’s (or earlier) where it was first used.
Authors’ response: We thanks the reviewer for this comment. There are of course many ways to correct the QT interval. This has been well studied in humans and lead to the famous Bazett equation that is used all over the world since 1920 (we have added this reference to our manuscript). However, we should also stress that Bazett’s formula is not adapted to zebrafish which is why we used Holzgrefes’ formula.
- Another problem with the manuscript is the comparison of QTc and RR. QTc is the QT interval corrected for the heart rate. Since heart rate is the RR interval in milliseconds, comparing QTc to RR must be linear, and makes not much sense to plot it.
Authors’ response: We agree with the reviewer that QTc-RR must be linear but this verification is rarely performed. Indeed, the best way to show that the correction formulae renders the QT independent of the RR interval, is to plot the QTc vs RR. If the QTc is independent of the RR, the relationship will be linear, there is no correlation. On the other hand, a positive or negative relationship indicates an incomplete correction or worst, a bias introduced by the correction. This is why we show, in figure 4C and D, the influence of the correction formulae on the QTC=f(RR) relationship.
- The electrode position determines whether the T wave has a positive or negative deflection (check for example Crowcombe et al 2016 (PlosOne)). It needs to be discussed.
Authors’ response: We thank the reviewer for this comment. However, as with Dillon et al, Crowcombe et al analysed larval ECG. In their study they positioned their electrode on the atrium, the AV canal or the ventricle which is very different to our own recordings in adult zebrafish. We have now included a schematic of our ECG electrode positioning which we hope will clarify this point.
- I assume that the RR intervals listed on the tables are in milliseconds not seconds.
Authors’ response: We thank the referee for this remark. We apologize for this mistake and have corrected this in the revised manuscript.
Throughout the manuscript reading I asked myself, why the authors recorded from adult zebrafish that required animal permit and seemingly made the recording difficult. It is much easier to do ECG recording from zebrafish larvae that needs no animal permit, can be done in medium to high throughput and pharma companies use it for drug discovery experiments rather than adult zebrafish.
Authors’ response: We thank the reviewer for this interesting comment. Indeed there are distinct advantages in using larvae, such as high throughput screening. The reason we chose to analyse adult zebrafish is because they have a fully developed heart/conduction system and as such are much closer to adult humans than a 3day old zebrafish larvae. Our goal is to establish a reliable model system for testing cardiotoxicity, not for the initial high throughput drug discovery. Probably the best way to envisage this is that the larvae would be used to identify potentially interesting compounds which would then be assessed in detail using adult zebrafish.
However, there are merits of the manuscript. For example, the experiment to gain insight to the effect of hypothermia ECG is worth expanding and further exploring, and I suggest taking the hypothermia experiment and publish it separately.
Authors’ response: We thank the referee for this comment and the suggestion. We are planning a more thorough study on the effects of hypothermia based on these preliminary results.
Reviewer 2 Report
In this manuscript, Arel et al. compared the three approaches (unexposed, exposed, and isolated heart) to record the electrocardiogram (ECG) performed on adult zebrafish and found that the exposed heart configuration is the most reliable strategy. They also showed that Holzgrefe’s formula is reliable in calculating the QTc value. Given that the ECG is the only and most widely used diagnostic tool to monitor heart activity in live zebrafish, the findings uncovered by this study are informative and valuable to the community. Specific comments were provided below on the areas that could be improved before it is published:
- The authors are encouraged to provide images or schematic diagrams showing how the three configurations (unexposed, exposed and isolated heart) were set up for ECG recording (e.g., how many and where the electrodes were placed). The appropriate citation should be given if any of the configurations have been described in previously published work. Given that recent studies cited (e.g., ref18, ref4) using the approach similar to the unexposed design in this study acquired detectable T waves, the authors are encouraged to discuss the possible reasons why T waves are hard to detect in their settings.
- The authors are encouraged to provide more details in M&M., e.g., line 126-127, Since the fish were subjected to surgeries to open the pericardial sac and expose the heart, how those fish were maintained to avoid infections around the wound and the heart, that may affect the following ECG recording on the extracted heart. Line 153-164, how the fish were treated upon low temperature and osmotic shocks before ECG performed on unexposed and exposed heart configuration? So far, only the isolated heart configuration was covered.
- Figures 2-3: It is difficult to see the distribution of “o” and “square” in the area overlay with “•”, please reformat the figure to better present the data; Please clarify what the sample size “n” indicates, as there are way more dots in figures than the “n” numbers indicate in legends; please describe the “ ∇ ” in Fig.2C.
- Please check the Fig.4 legend to make the description match the presentation shown in the figure.
- Table legend: the authors are encouraged to provide the total number of fish used in each configuration condition; Please clarify what the symbols “**,*****,$$” indicate; please describe how the SEM values were obtained in supplemental tables.
- Typos: line 153-164, “CaCl2” should be “CaCl2”. Line 246, please clarify what effects were reversible.
Author Response
Reviewer 2
In this manuscript, Arel et al. compared the three approaches (unexposed, exposed, and isolated heart) to record the electrocardiogram (ECG) performed on adult zebrafish and found that the exposed heart configuration is the most reliable strategy. They also showed that Holzgrefe’s formula is reliable in calculating the QTc value. Given that the ECG is the only and most widely used diagnostic tool to monitor heart activity in live zebrafish, the findings uncovered by this study are informative and valuable to the community. Specific comments were provided below on the areas that could be improved before it is published:
- The authors are encouraged to provide images or schematic diagrams showing how the three configurations (unexposed, exposed and isolated heart) were set up for ECG recording (e.g., how many and where the electrodes were placed). The appropriate citation should be given if any of the configurations have been described in previously published work. Given that recent studies cited (e.g., ref18, ref4) using the approach similar to the unexposed design in this study acquired detectable T waves, the authors are encouraged to discuss the possible reasons why T waves are hard to detect in their settings.
Authors’ response: We thank the referee for their encouraging comments. A schematic diagram has been added in figure 1 as suggested. Indeed, our data are in line with the recent study by Liu et al (ref.4) which shows that removing the dermis and exposing the heart results in much clearer T waves (we have now commented on this in the discussion). The ECG recording system used by Le et al (ref 18) is custom built and very different from our standard (and widely available) ECG system. It would be very difficult for someone wishing to record zebrafish ECG to set up the Le et al system.
- The authors are encouraged to provide more details in M&M., e.g., line 126-127, Since the fish were subjected to surgeries to open the pericardial sac and expose the heart, how those fish were maintained to avoid infections around the wound and the heart, that may affect the following ECG recording on the extracted heart. Line 153-164, how the fish were treated upon low temperature and osmotic shocks before ECG performed on unexposed and exposed heart configuration? So far, only the isolated heart configuration was covered.
Authors’ response: The referee is perfectly right. No treatment was applied to the fish or in the water after surgery to avoid affecting the ECG recording. We followed strict rules regarding the wellbeing of the animals and exclusion of the experiment and euthanasia was planned if the fish showed any sign of infection or distress. Despite these measures, no exclusion was necessary as no sign of infection or distress were observed. We have included this information in the M&M section, ECG recording: “No treatment was applied after surgery in order to avoid affecting the ECG recordings.”
We have also described how the osmotic shock and hypothermic solutions were applied in both the unexposed and exposed heart configurations in the M&M section: “For both the unexposed and exposed heart configurations we first removed 1mL of the bath solution, then 1mL of the osmotic shock solution or 1ml of the hypothermic solution was applied onto the cardiac area prior to measurement. ”
- Figures 2-3: It is difficult to see the distribution of “o” and “square” in the area overlay with “•”, please reformat the figure to better present the data; Please clarify what the sample size “n” indicates, as there are way more dots in figures than the “n” numbers indicate in legends; please describe the “ ∇ ” in Fig.2C.
Authors’ response: We apologise that our figure was not clear. We have now changed the point format and added details regarding the sample size in the legends and the figure panels. Briefly, each symbol represents a single QT measurement and its associated RR interval. These measurements were obtained from groups of 3 to 10 fish (details for each graph are provided in the figure legends). As for Fig.2C, we have corrected the mismatched symbols.
- Please check the Fig.4 legend to make the description match the presentation shown in the figure.
Authors’ response: We apologise to the reviewer for this mistake. We have now corrected the Fig.4 legend in the manuscript.
- Table legend: the authors are encouraged to provide the total number of fish used in each configuration condition; Please clarify what the symbols “**,*****,$$” indicate; please describe how the SEM values were obtained in supplemental tables.
Authors’ response: We apologize for the lack of clarity regarding the symbols we used to denote significance. We have now ensured this is uniform throughout the manuscript and complies the accepted norms, *P<0.05, **P<0.01, ***P<0.001, ****P<0.0001. We have also provided more detailed information about the statistical analyses in the M&M section. Regarding the SEM of the supplemental tables, it was calculated using GraphPad Prism 9.2.0. We have now included this information in the M&M section.
- Typos: line 153-164, “CaCl2” should be “CaCl2”. Line 246, please clarify what effects were reversible.
Authors’ response: We thank the reviewer for noticing this typo which we have corrected. Regarding the effects of hypothermia, we considered that the effects of hypothermia were reversible when the warm washout conditions were not statistically different from the warm control conditions. We have now included this information in the manuscript ‘No significant differences were observed between the warm (control) and the warm (washout) conditions indicating that these effects are all reversible’
Reviewer 3 Report
This manuscript by Arel et al characterized cardiac toxicity by performing electrocardiogram in adult zebrafish with the ultimate goal of robust drug development. This manuscript, in general, is well-written and the current study potentially warrant further investigation. Nevertheless, this manuscript could be further strengthened by addressing comments blow.
- Introduction is well-written. Anatomical and pharmacological cardiac toxicity and disturbance in QT interval is well known. Although description of QRS complex and history of zebrafish ECG measurement is crucial for the readership, it is unnecessarily long. Please reduce the context or supplement the information in the discussion section.
- Indeed zebrafish emerged as critical developmental model, Please supplement the last paragraph of the introduction with clinical significance of this work.
- The title does not entirely reflect the data. The title will benefit from an elaboration; it needs to be more direct.
- What was to criteria to evaluate the presented characterization is precise? I believe the current data lacks controls and a ground truth for characterization. Parallel studies such as optical mapping of conduction must be performed.
- Simple husbandry and rapid development are major advantages of using zebrafish. However, studies in Figure 1 and Table 1 were performed with a single subject. My humble opinion is this dilute overall quality of data. Please consider increasing the sample number.
- Authors described that the ultimate goal of this study is to facilitate high-throughput drug development. In order to claim zebrafish ECG model, authors must supplement the data or at least discuss pharmacological administration to induce cardiac toxicity, and seek whether drug(s) rescues the effect. The current literature introduces chemotherapy (Doxorubicin)-induced dilated cardiomyopathy in zebrafish models. Potentially doxorubicin is a good target to start with.
- Please elucidate the number of measurements in figure legends (Figure 2-3)
- The readership may benefit from introduction of the ECG system as a data supplement.

Author Response
Reviewer 3
This manuscript by Arel et al characterized cardiac toxicity by performing electrocardiogram in adult zebrafish with the ultimate goal of robust drug development. This manuscript, in general, is well-written and the current study potentially warrant further investigation. Nevertheless, this manuscript could be further strengthened by addressing comments blow.
- Introduction is well-written. Anatomical and pharmacological cardiac toxicity and disturbance in QT interval is well known. Although description of QRS complex and history of zebrafish ECG measurement is crucial for the readership, it is unnecessarily long. Please reduce the context or supplement the information in the discussion section.
Authors’ response: We thank the reviewer for this comment. We have edited the introduction along the lines suggested by the reviewer and moved the part of the introduction describing the history of zebrafish ECG measurement to the discussion section.
- Indeed zebrafish emerged as critical developmental model, Please supplement the last paragraph of the introduction with clinical significance of this work.
Authors’ response: We thank the reviewer for this comment. We have now developed the last paragraph of the introduction as suggested by the reviewer: “To this end, we also sought to determine what effect clinical therapeutic treatments such as hyperosmolality (employed during traumatic brain injury treatement) and hypothermia/ targeted temperature management (employed during stroke/heart attack treatment) had on ECG recordings in the 3 different configurations. Because these treatments are often combined with medication, we sought to determine whether adult zebrafish can be used as a screening model for novel pharmaceuticals which could be utilised in combination with these therapeutic treatments. ”
- The title does not entirely reflect the data. The title will benefit from an elaboration; it needs to be more direct.
Authors’ response: We agree with the reviewer and have now changed the title of the manuscript: “The effect of hypothermia and osmotic shock on the electrocardiogram of adult zebrafish.”
- What was to criteria to evaluate the presented characterization is precise? I believe the current data lacks controls and a ground truth for characterization. Parallel studies such as optical mapping of conduction must be performed.
Authors’ response: We apologise to the reviewer as we do not fully understand this comment. The controls for the osmotic shock experiments are the isosmotic conditions and the controls for the hypothermia experiments are the warm conditions. Regarding the characterization of conduction in zebrafish this has been performed recently by Zhao et al. (2021). We thus do not think that such experiments will particularly strengthen the take-home message of this paper, even if we agree that it is another way to support our conclusions. Furthermore, as far as we are aware optical mapping can only be performed on larvae or on isolated hearts. In our study we suggest the exposed heart configuration is optimal to record ECG, and since this configuration cannot be tested using optical mapping, we believe it would not be suited.
- Simple husbandry and rapid development are major advantages of using zebrafish. However, studies in Figure 1 and Table 1 were performed with a single subject. My humble opinion is this dilute overall quality of data. Please consider increasing the sample number.
Authors’ response: We apologize for the lack of clarity. Figure 1 is a descriptive figure showing representative traces recorded in the different configurations. Table 1 however, was built by analysing data recorded from up to 20 fish. In order to clarify this point, we have indicated the sample size in all tables.
- Authors described that the ultimate goal of this study is to facilitate high-throughput drug development. In order to claim zebrafish ECG model, authors must supplement the data or at least discuss pharmacological administration to induce cardiac toxicity, and seek whether drug(s) rescues the effect. The current literature introduces chemotherapy (Doxorubicin)-induced dilated cardiomyopathy in zebrafish models. Potentially doxorubicin is a good target to start with.
Authors’ response: We apologise to the reviewer if we have not been clear, but our goal was not to facilitate high-throughput drug development. As mentioned in the response to reviewer 1, larvae are much more suited to high throughput screening. The reason we chose to analyse adult zebrafish is because they have a fully developed heart/conduction system and as such are much closer to adult humans than a 3day old zebrafish larvae. Our goal is to establish a reliable model system for testing cardiotoxicity, not for the initial high throughput drug discovery. Probably the best way to envisage this is that the larvae would be used to identify potentially interesting compounds which would then be assessed in detail using adult zebrafish. The proposed experimental plan to identify drugs which could rescue Doxorubicin induced cardiomyopathy is an excellent idea but would be much better suited to using larval zebrafish and is beyond the scope of our manuscript.
- Please elucidate the number of measurements in figure legends (Figure 2-3)
Authors’ response: We apologise to the reviewer that the number of measurements was not clearly indicated. This has been addressed and the number of measurements and sample sizes are inidicated more clearly in all figures and their legends.
- The readership may benefit from introduction of the ECG system as a data supplement.
Authors’ response: We thank the reviewer for this insightful comment. We have now included a figure detailing the ECG equipment and how this was set up to record adult zebrafish ECG (supplemental Figure 1).